# A Sentinel Serological Study in Selected Zoo Animals to Assess Early Detection of West Nile and Usutu Virus Circulation in Slovenia

**DOI:** 10.3390/v13040626

**Published:** 2021-04-06

**Authors:** Pavel Kvapil, Joško Račnik, Marjan Kastelic, Eva Bártová, Miša Korva, Mateja Jelovšek, Tatjana Avšič-Županc

**Affiliations:** 1Veterinary Department, Ljubljana Zoo, Večna Pot 70, 1000 Ljubljana, Slovenia; marjan.kastelic@zoo.si; 2Department of Biology and Wildlife Diseases, Faculty of Veterinary Hygiene and Ecology, University of Veterinary Sciences Brno, Palackého tř. 1946/1, 612 42 Brno, Czech Republic; bartovae@vfu.cz; 3Institute for Poultry, Birds, Small Mammals, and Reptiles, Faculty of Veterinary Medicine, University of Ljubljana, Gerbičeva 60, 1000 Ljubljana, Slovenia; josko.racnik@vf.uni-lj.si; 4Institute of Microbiology and Immunology, Faculty of Medicine, University of Ljubljana, Zaloška 4, 1000 Ljubljana, Slovenia; Misa.Korva@mf.uni-lj.si (M.K.); Mateja.Jelovsek@mf.uni-lj.si (M.J.); Tatjana.Avsic@mf.uni-lj.si (T.A.-Ž.)

**Keywords:** infectious diseases, zoo environment, West Nile virus, Usutu virus epidemiology, biosurveillance

## Abstract

Monitoring infectious diseases is a crucial part of preventive veterinary medicine in zoological collections. This zoo environment contains a great variety of animal species that are in contact with wildlife species as a potential source of infectious diseases. Wild birds may be a source of West Nile virus (WNV) and Usutu (USUV) virus, which are both emerging pathogens of rising concern. The aim of this study was to use zoo animals as sentinels for the early detection of WNV and USUV in Slovenia. In total, 501 sera from 261 animals of 84 animal species (including birds, rodents, lagomorphs, carnivores, ungulates, reptiles, equids, and primates) collected for 17 years (2002–2018) were tested for antibodies to WNV and USUV. Antibodies to WNV were detected by indirect immunofluorescence tests in 16 (6.1%) of 261 animals representing 10 species, which were sampled prior to the first active cases of WNV described in 2018 in Slovenia in humans, a horse, and a hooded crow (*Corvus cornix*). Antibodies to USUV were detected in 14 out of 261 animals tested (5.4%) that were positive prior to the first positive cases of USUV infection in common blackbirds (*Turdus merula*) in Slovenia. The study illustrates the value of zoological collections as a predictor of future emerging diseases.

## 1. Introduction

West Nile virus (WNV) and Usutu virus (USUV), in the family *Flaviviridae*, are emerging enzootic pathogens with epizootic and epidemic potential in Europe. Their transmission cycle involves many mosquito species as vectors (mainly *Culex* species) and birds as the primary reservoir host of these viruses [1]. Incidental hosts such as horses, humans, other mammals may be infected, resulting in febrile illness, meningitis, encephalitis, and, in the worst case, a fatal outcome. When birds are the reservoir for the virus, they usually appear healthy, but they are infected and therefore carrying the virus, perhaps even for long distances to new geographic regions during migration. On the other hand, birds that become infected may develop symptoms arising from the infection. The first positive active cases of WNV in people in Slovenia were detected in 2018 in three patients with IgM antibodies in the cerebrospinal fluid of the patients [2]. In animals, the first active cases of WNV in Slovenia were detected in a hooded crow (*Corvus cornix*) and a horse in the Ljubljana region in 2018 [3]. In the same year, the first cases of USUV in Slovenia were detected in a common blackbird (*Turdus merula*) and a song thrush (*Turdus philomelos*) [4]. Based on the successful virus isolation and complete genome sequencing, connection between positive human cases and WNV found in mosquitos has been made [2]. Additionally, the whole genome sequence indicates that virus most likely spread from the north because of the geographic proximity as well as the sequence cluster with the Austrian and Hungarian sequences [2]. Captive and free-ranging animals have been used quite often as sentinels to monitor arbovirus infections in the past decades [5,6]. Animals from the zoological collections are not commonly used for antigen screening to evaluate the epidemiological situation of a disease in the wider area. However, the routine monitoring of wild crows found dead in the Bronx Zoo in New York contributed to the discovery of WNV as a cause of human morbidity and mortality in the United States in 1999 [7,8,9,10]. In that context, samples from zoo animals collected during routine procedures or so-called biobanking could be important for retrospective and prospective examination of samples for possible emerging pathogens and for monitoring the epizootiological situation from a long-term perspective [11]. Animals in zoological collections can provide an excellent observation group due to the variety of animal species and their high susceptibility to different pathogens such as WNV and USUV. Zoo animals are also in direct or indirect contact with wild birds and arthropods, which are potential carriers of WNV and USUV.

This study analyzes samples of sera taken from zoo animals during a 17-year period (2002–2018), using zoo animals as sentinels to investigate whether antibodies to WNV and USUV were present in animals from the Ljubljana Zoo before the infections were detected in a horse, a crow, a blackbird, and humans in Slovenia.

## 2. Materials and Methods

### 2.1. Animals

The study took place at the Ljubljana Zoo in Slovenia, which has a large collection of exotic animal species. The zoo covers a hilly area of 20 hectares, which is heavily forested and surrounded by wetlands.

Blood samples were collected from zoo animals during routine clinical procedures such as annual health checks of animals and elective diagnostic and therapeutic procedures or surgeries. Before full-body clinical examination, the animals were manually or chemically restrained. Blood was collected from the most appropriate vein based on the animal species. Altogether, 501 blood samples from 261 animals of 84 animal species including birds (*n* = 284), rodents (*n* = 80), lagomorphs (*n* = 5), carnivores (*n* = 18), ungulates (*n* = 72), reptiles (*n* = 9), equids (*n* = 5), primates (*n* = 26), and marsupials (*n* = 2) were collected over a period of 17 years (2002–2018).

Samples of sera were divided into three groups according to periods of sampling (period 1 from 2002 to 2013, period 2 from 2014 to 2015, and period 3 from 2016 to 2019). The sampling periods were selected based on the retrospective importance regarding emerging WNV and USUV spreading across Europe. The purpose of defining these periods was to simplify the presentation of the results and highlight the occurrence of both emerging diseases.

### 2.2. Serological Methods

The sera of animals were tested for antibodies to WNV and USUV by using an indirect immunofluorescence assay (IFA) using species-specific or species-related conjugates based on the animals tested (Bethyl Laboratories Inc., Montgomery, TX, USA). The conjugates used in the study were anti-chicken IgG (*n* = 44), anti-bird IgG (*n* = 249), anti-dog IgG (*n* = 9), anti-cat IgG (*n* = 9), anti-bovine IgG (*n* = 9), anti-goat IgG (*n* = 1), anti-sheep IgG (*n* = 8), anti-deer IgG (*n* = 50), anti-horse IgG (*n* = 8), anti-pig IgG (*n* = 3), anti-guinea pig IgG (*n* = 32), anti-rabbit IgG (*n* = 32), anti-mouse IgG (*n* = 34), anti-rat IgG (*n* = 14), and anti-monkey IgG (*n* = 26) (Appendix A). The IFA and serum neutralization test (SNT) methods was performed as formerly described by Knap et al. 2020 [11].

IFA positive samples have been tested for antibodies to Tick-borne encephalitis virus (TBEV) with enzyme-linked immunosorbent assays (ELISA, EIA TBEV Ig, TestLine Clinical Diagnostic, Brno, Czech Republic), using its crossreaction with other flaviviruses. Positive samples were confirmed for TBEV (Strain Hypr), WNV (WNV Strain Line 2), and USUV by a virus neutralization test (VNT) in a micromodification format, with vital staining (7 CV-1 cell suspension, monkey kidney cell line) used as a cell substrate for WNV and USUV, and suspension of porcine kidney cell line (PS cells) used as a cell substrate for TBEV, with a working dilution of 600,000 cells/mL for both cell lines. The result of VNT is a virus neutralization (VN) titer, which is the reciprocal of the highest sample dilution that is still capable of neutralizing the cytopathic effect, due to the WNV and/or USUV in at least, 50% of each monolayer. The samples were scored as positive if the VN titer exceeded the dilution of 1:4.

## 3. Results

Altogether, antibodies to WNV were detected by IFA in 16 (6.1%) of 261 animals from 10 different species; nine of them were mammals (3.5%) and seven were birds (2.9%). In the case of USUV, antibodies were detected by IFA in 14 of 261 animals (5.4%); six of them were birds (2.2%) and eight were mammals (3.1%). All serologically positive animals were clinically healthy, and no signs of disease have been recorded for 24 months since the last sampling.

In WNV-positive animals, 10 were tested only once: one wild rabbit (*Oryctolagus cuniculus*), two Eurasian wolves (*Canis lupus lupus*), and two northwestern wolves (*Canis lupus occidentalis*) tested in 2017, and one guinea pig (*Cavia porcellus*), one Patagonian mara (*Dolichotis patagonum*), one wild boar (*Sus scrofa*), and two red foxes (*Vulpes vulpes*) tested in 2018. Six seropositive animals were tested more than once. One helmeted guineafowl (*Numida meleagris*) was tested in 2015 and 2017, two eastern white pelicans (*Pelecanus onocrotalus*), one tested in 2015 and 2017, and the other tested in 2015, 2017, and 2018, one guinea pig (*Cavia porcellus*) tested twice in 2018, one Eurasian eagle owl (*Bubo bubo*) tested in 2017 and twice in 2018, one barn owl (*Tyto alba*) tested in 2015, 2017, and 2018, and one snowy owl (*Nyctea scandiaca*) tested twice in 2018. In USUV-positive animals, seven were tested once. One wild rabbit, two Eurasian wolves, and one northwestern wolf were tested in 2017, and a guinea pig, Patagonian mara, and red fox were tested in 2018. Seven seropositive animals were tested more than once. Animals tested on multiple occasions included two eastern white pelicans, one of them tested in 2015 and 2017 and the other tested in 2015, 2017, and 2018, one guinea pig tested twice in 2018, two Eurasian eagle owls tested in 2017 and twice in 2018, one barn owl tested in 2015, 2017, and 2018, and one snowy owl tested twice in 2018.

All the samples were negative for WNV or USUV prior to 2016, and all the animals were negative at least 1 year prior to the positive result. All positive samples were collected prior to 2018, when the first cases of WNV in humans, a horse, and a hooded crow and USUV in a blackbird were detected in Slovenia. A barn owl sampled in 2015, 2017, and 2018 tested positive in 2018 by IFA and VNT with a titer of 128 for both viruses (WNV and USUV). Another 15 IFA-positive animals tested negative with VNT. One sample was too small for evaluation and was not tested with VNT. A summary and characteristics of the animals that tested positive is provided in Table 1.

## 4. Discussion

The epidemiological situation of WNV has gone through dramatic changes during the past 30 years. Outbreaks of WNV in people and animals have been reported across Europe since 1996. One of the largest outbreaks was reported in Romania in 1996 and 1997, with more than 500 clinical cases in humans with almost 10% mortality [12]. Since 2015, annual WNV outbreaks of varying intensities in wild animals have been reported in France [13]. In 2018, a dramatic increase in WNV infections in people and animals was reported across Europe, with the most cases in Italy, Serbia, and Greece [14]. In Europe, there were 2083 cases of WNV outbreaks in humans and 285 outbreaks among equids from 1999 to 2019 [14]. Initially, WNV lineage 1 was identified as a cause of human outbreaks across Europe, whereas WNV lineage 2 was introduced in Europe later, in 2004, and then dispersed to eastern Austria and other European countries as a major prevalent strain [13]. In 2018, WNV lineage 2 appeared in southeastern Europe. This virus lineage 2 most probably originated in northern Italy and caused West Nile neuro-invasive disease in humans and the death of diurnal raptors [13]. WNV lineage 2 was also responsible for positive cases in Slovenia [15].

In Slovenia, three cases of WNV infection were confirmed by the detection of IgM antibodies in the cerebrospinal fluid of humans in 2018 [14]. Retrospective examination of samples from wild bird carcasses collected from 2010 to 2012 did not provide evidence of WNV [16]. However, 4.7% of serum samples from 34 different species of songbirds, including resident species, tested from 2003 to 2009 were positive for antibodies to WNV [17]. In our study, the overall prevalence was 2.9% in birds, with no sampled animals developing clinical signs or a fatal outcome. Comparable results were obtained in southern Spain, where antibodies to WNV were detected in 7.8% of 142 feral pigeons and in 8.2% of 49 zoo animals [18].

Usutu virus is another emerging arbovirus with outbreaks reported in southern Europe since 2001 [18,19]. This virus was first discovered in Africa in 1959 and subsequently for the first time on record, in southern Europe (Austria) in 2001, before gradually dispersing throughout many parts of mainland Europe. Moreover, WNV and USUV-specific neutralizing antibodies were detected in a wide range of healthy wild migratory and non-migratory birds, and sentinel chickens, in England, as reported in 2003 and 2006 [20,21]. The African lineage of Usutu virus has now been isolated from birds in the UK in 2020 [22]. It is assumed that this USUV was introduced from Belgium or Holland where the same strain has been detected recently.

This infectious agent seems to co-circulate with WNV [23]. In Europe, USUV has been shown to be highly pathogenic for several bird species, especially blackbirds and great gray owls (*Strix nebulosa*) [23]. Furthermore, neurotropism of USUV for humans was reported for the first time in both immunocompromised and immunocompetent patients [14]. USUV has been directly detected in bats, and antibodies to USUV have been detected in various animal species (horses, dogs, squirrels, wild boar, deer, and lizards [24]. In our study, both seropositive mammals and birds were without any clinical signs of disease. This result could be expected in mammals used as a sentinel; however, in the case of birds (especially in owls), the absence of clinical disease is surprising. Although USUV is known to cause clinical disease and death in susceptible birds, such as owls, all four positive owls in our study were asymptomatic. One of two positive Eurasian eagle owls showed a very low titer of 10 and tested negative twice over the course of the following year. However, barn owls were positive for antibodies for WNV as well as for USUV. It seems that there was simultaneous coinfection with both viruses because there was a high VNT titer of 1:128 for both viruses. Moreover, the positive sample was collected in autumn 2018, which was the year when both viruses (WNV and USUV) were introduced into Slovenia, with the first confirmed cases in humans and animals. The barn owl showed no clinical signs and was still alive 24 months after the positive sample collection.

However, an interesting question is why IFA tests were positive but only one VNT test produced a positive result. One possibility might be that since these animals have benefitted from being kept under the protective environment of the zoo, including the regular provision of food, protection from predators, and climatically harsh conditions, and, as they also have records of having been healthy for years, they clearly did not suffer severe infections due to WNV or USUV. This might explain why low levels of virus-specific antibody could be detected in their sera (i.e., due to mild or asymptomatic infections), but only in one case was there evidence of a potent neutralizing response (and therefore maybe a symptomatic infection occurred), but in the protective environment of the zoo, the bird was sufficiently robust to recover from the infection).

Horses, dogs, and other domestic mammals were used as mammal sentinels of WNV infections, in which dogs showed promising potential [15,20,25]. The main advantage of the captive sentinel is good monitoring of the animal and the ability to supervise collection and contact with other animals. Captive birds, which were typically chickens or pigeons positioned in suspected loci of the disease, were often used as bird sentinels of WNV infection [5,6,21,26]. Free-ranging sentinels in the case of WNV are usually wild birds, which are captured, blood-sampled, and released. The possibility of a dynamic model of samples is secured by banding or marking the bird. They have been widely used in monitoring arbovirus infections [5,16] or other pathogens [27]. A huge advantage of wild living animals is their ability to roam freely and cover large areas, especially in the case of birds. This increases the possibility of interaction with infected mosquitos and the probability of spending some time in an arbovirus-positive geographical area. Unfortunately, during recapture, researchers are not able to trace the previous movement of the animal and track the enzootic focus. However, a new dynamic model could be offered by a combination of free-ranging birds as sentinels and zoological collections as epidemiological stations. A study of wild birds from 2013 by Račnik found zero prevalence of WNV RNA in free-ranging birds, which corresponds to the same results (zero prevalence in 2002–2013 and 2014–2015) in serological samples obtained during that time from zoo animals in our study [16].

Zoological collections as sentinels offer a higher variety of different animal species usually covering a larger area with more natural or artificial biotypes. There have been few important studies focusing on the potential of zoo animals as a sentinel for detection of zoonotic flaviviruses. After the outbreak of WNV in New York City in 1999, monitoring of animals in the zoological collection at the Bronx Zoo and Wildlife Conservation Park (BZ/WCP) showed that 34% of birds in the zoo were positive for WNV antibodies, 22% showed signs of disease, and 70% of diseased birds died [10]. On the other hand, only 8% of mammals in the zoo were positive for WNV antibodies, with zero mortality. The other study was carried out in Spain at 10 zoos between 2002 and 2019, where antigenically-related flavivirus antibodies were detected by ELISA in 3.3% of 570 zoo animals belonging to 10 (8.3%) of 120 mammal species [28]. These results demonstrated that zoo mammals were exposed to WNV, USUV, and TBEV infection, and that they can be used as sentinels for emerging pathogens. However, this study was carried out in an area where WNV was already an endemic disease. In our study, we have demonstrated the role of zoo animals as a possible early warning system for pathogens infiltrating into new animal populations and infecting immunologically naive animals. Zoo animals were also used for detecting USUV infection. Antibodies to USUV were detected in 8.8%, 5.3%, and 6.6% of 372 birds sampled between 2006 and 2007 in four zoological collections in Austria, Switzerland, and Hungary, respectively [28]. These numbers correspond to the prevalence found in our study (5.4%). The authors of this study recommended regular testing of zoo birds for WNV and USUV infection, which might prove especially useful in naive populations. In another very recent study from France, 411 samples from 70 species were collected over 16 years from 2003 to 2019 [29]. Surprisingly, USUV seroprevalence in birds in this study was 10 times higher than that of WNV (14.59% vs. 1.46%, respectively). Infections occurred mainly between 2016 and 2018, corresponding to the virus circulation in Europe and its presence in 2018 in Slovenia.

Serological screening of different zoo animal species might provide an early warning system for detecting pathogens such as WNV and USUV. Zoological collections in the role of an epidemiological station could play an important role in the early detection of emerging pathogens and their prevention in the future.

## Figures and Tables

**Table 1 viruses-13-00626-t001:** Characteristics of zoo animals positive for West Nile virus (WNV) and Usutu virus (USUV) with dynamic of titers in indirect immunofluorescence assay (IFA) and virus neutralization test (VNT) by years of sampling.

Species	Latin Name	Borned	Sex	Date of Sampling	IgG	WNV IFA	USUV IFA	WNV VNT	USUV VNT
Guineafowl	*Numididae* *meleagris*	June 2009	M	2015	anti-chicken/bird	neg	neg	neg	neg
				May 2017		40	neg		
Pelican	*Pelecanus* *onocrotalus*	June 1987	M	2015	anti-bird	neg	neg		
				December 2016		80	neg	neg	neg
				September 2018					
				December 2019		40	40		
Pelican	*Pelecanus* *onocrotalus*	January 1998	F	2015	anti-bird	neg	neg		
				December 2016		neg	40	neg	neg
Eurasian eagle-owl	*Bubo bubo*	April 2004	M	December 2017	anti-bird	neg	10		
				September 2018		neg	neg	neg	neg
				November 2018		neg	neg		
Eurasian eagle-owl	*Bubo bubo*	June 1990	M	December 2017	anti-bird	neg	neg		
				September 2018		neg	neg	neg	neg
				November 2018		40	40		
Barn owl	*Tyto* *alba*	June 2000	F	2015	anti-bird	neg	neg		
				December 2017		neg	neg		
				Sep 2018		40	10	128	128
Snowy owl	*Nyctea* *scandiaca*	June 1996	F	October 2018	anti-bird	40	neg		
				November 2018		10	10		
Guinea pig	*Cavia* *porcellus*	March 2014	F	November 2017	anti-guinea pig	neg	neg	neg	neg
				October 2018		10	20		
Rabbit	*Oryctolagus cuniculus*	June 2012	F	May 2017	anti-rabbit	20	10	neg	neg
Gray wolf	*Canis* *lupus*	May 2002	M	April 2017	anti-dog	10	40	neg	neg
Northwestern wolf	*Canis* *lupus occidentalis*	April 2016	F	April 2017	anti-dog	80	80	neg	neg
Northwestern wolf	*Canis* *lupus occidentalis*	April 2016	F	April 2017	anti-dog	40	neg	neg	neg
Gray wolf	*Canis* *lupus*	May 2002	M	April 2017	anti-dog	10	10	neg	neg
Guinea pig	*Cavia* *porcellus*	young	F	June 2018	anti-guinea pig	neg	40		
Patagonian mara	*Dolichotis* *patagonum*	March 2017	F	January 2018	anti-guinea pig	neg	40	neg	neg
Wild boar	*Sus scrofa*	March 2017	M	November 2017	anti-pig	40	neg	neg	neg
Red fox	*Vulpes vulpes*	young	F	August 2018	anti-dog	10	neg	s.s.	s.s.
Red fox	*Vulpes vulpes*	young	M	November 2018	anti-dog	>160	>160	neg	neg

M = male, F = female, neg = negative, s.s. = small sample.

## Data Availability

The data that support the findings of this study are available on request from the corresponding author. The data are not publicly available due to privacy or ethical restrictions.

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
