# Peer review of "A Sentinel Serological Study in Selected Zoo Animals to Assess Early Detection of West Nile and Usutu Virus Circulation in Slovenia"

_viruses, 2021, doi:10.3390/v13040626_

Round 1

Reviewer 1 Report

Dear Authors,

the presented topic for the examination of zoo animals is important for early detection of emerging diseases. Although the reviewer agrees with the general design of the study and with the methods used, many critical points should be revised.

A thorough revision of the following points should be carried out:

The correct abbreviation for Usutu virus is USUV and not UV. This error continues through the entire manuscript and should be revised.

Introduction:

  • Why are the important surveillance studies with the results and the human WNV infection are not involved here, as described in the discussion part or in the paper by Knap et al. 2020 about WNV in Slovenia ?
  • What about the WNV and USUV situation in the relevant neighboring countries as a potential introduction route? This should be implemented.

Material and Methods:

  • An Overview about the relevant mammal and bird families and the numbers investigated per year is helpful for the readers to better understand the manuscript as a longitudinal study over 17 years. This should be implemented as a well-arranged table in the appendix (see by Constant et al. 2020, https://doi.org/10.3390/pathogens9121005) or as a representative figure with pie charts or bar graphs
  • the method descriptions are too superficial, the exact details on how to perform the IFA are completely missing, only the used conjugate was included, but is this anti-bird IgG (correct IgY) species-specific enough for the testing of different bird species (owls and others) ?
  • furthermore, the correct description of this antigen for birds must be “IgY” and not IgG !
  • the presentation of the SNT method is more than superficial. No reference is made whether this is an in-house method or a reference to an already published method. And the description about the possibility of cross-reaction between USUV and WNV in both methods is completely missing. For a better description of the methods see the publication of Knap et al. 2020 or Constant et al. 2020
  • Why you describe the TBEV in the procedure of the VNT method? Do you use this virus also to specify the antibody reaction in mammals?

Results:

  • The table presented is not clear and contains errors. Because, by VNT the neutralizing immune response was considered specific if the VNT titer for any given virus was at least fourfold higher than titers obtained against the other viruses. And if the antibody titers for both viruses (WNV and USUV) were the same or differed only slightly (1 to 1,5 fold), it is not possible to discriminate between WNV or USUV titers and the result must be interpreted as not analyzable. Therefore e.g. for a pelican, a barn owl, a snowy owl, a wolf and a red fox the exact specific antibody reaction cannot be determined at the same time. Hence the presentation of the positive results must be revised. Or what are your comments when you have equal serological titers for WNV and USUV in same birds and mammals? What is with cross-reactions by the IFA and their results?

Discussion:

  • The part is quite well done and ties into many other results from zoo studies already.
  • The discussion should be adapted to the revised serological results and it would also be recommended to insert the zoo study from France (Constant et al. 2020)

Finally, the results of the study should be conveyed in a more attractive style with informative tables and figures of the results for a better character of this longitudinal study of zoo animals over 17 years as an important factor that zoological collections can use for early detection of emerging diseases.

Furthermore, after a major revision of all critical points and minding the fact that only a few methods were performed and the novelty of the results is limited, it should be rethought to publish this manuscript as a short communication form.

Minor points: In the supplemental part each scientific name of the bird and mammal species should be written in italics.

Author Response

Cover letter

A Sentinel Serological Study in Selected Zoo Animals to Assess Early Detection of West Nile and Usutu Virus Circulation in Slovenia

Dear editor,

Thank You very much for Your review, I do appreciate Your time and patience very much.

Pavel Kvapil

Please find the rseponse bellow

Reviewer 1

The correct abbreviation for Usutu virus is USUV and not UV. This error continues through the entire manuscript and should be revised.

USUV instead of UV has been replaced through the entire manuscript

Introduction:

  • Why are the important surveillance studies with the results and the human WNV infection are not involved here, as described in the discussion part or in the paper by Knap et al. 2020 about WNV in Slovenia?

Surveillance information study by Knap has been added into the introduction.

  • What about the WNV and USUV situation in the relevant neighboring countries as a potential introduction route? This should be implemented.

WNV and USUV potential of introduction has been included in the introduction.

Material and Methods:

  • An Overview about the relevant mammal and bird families and the numbers investigated per year is helpful for the readers to better understand the manuscript as a longitudinal study over 17 years. This should be implemented as a well-arranged table in the appendix (see by Constant et al. 2020, https://doi.org/10.3390/pathogens9121005) or as a representative figure with pie charts or bar graphs

I have not found any tables for materials and methods section in Constant et al 2020 just very detailed tables regarding results, where longitudinal view could be overshadowed by very precise table with all animals tested. We have tried to choose a different approach with simplified positive animals table and detailed overview in supplemental material.

  • the method descriptions are too superficial, the exact details on how to perform the IFA are completely missing, only the used conjugate was included, but is this anti-bird IgG (correct IgY) species-specific enough for the testing of different bird species (owls and others)?

For IFA the same lab as in Knap et al manuscript has been used with the same team and the similar description as for dog sera part in material and methods, anti bird conjugate commercial name is indeed IgG- (adding the link bellow)

https://www.bethyl.com/product/A140-110P/Bird+IgG-heavy+and+light+chain+Antibody_Seconary+Antibody

For additional information I am attaching the reference

Ebel, G. D., Dupuis, A. P., II, D. N., Young, D., Maffei, J., & Kramer, L. D. (2002). Detection by enzyme-linked immunosorbent assay of antibodies to West Nile virus in birds. Emerging infectious diseases8(9), 979.

  • furthermore, the correct description of this antigen for birds must be “IgY” and not IgG !

The same response as the remark above, anti bird  conjugate commercial name is indeed IgG- (adding the link bellow)

https://www.bethyl.com/product/A140-110P/Bird+IgG-heavy+and+light+chain+Antibody_Seconary+Antibody

  • the presentation of the SNT method is more than superficial. No reference is made whether this is an in-house method or a reference to an already published method. And the description about the possibility of cross-reaction between USUV and WNV in both methods is completely missing. For a better description of the methods see the publication of Knap et al. 2020 or Constant et al. 2020

As we did not present the SNT method I am not sure which part shall I respond to. But regarding Knap at al , we have used the same lab and the same team so the part of that corresponding woith our study has been described in a similar way. If this information is insufficient I can further contact the head of the Lab in order to provide the specific information .

  • Why you describe the TBEV in the procedure of the VNT method? Do you use this virus also to specify the antibody reaction in mammals?

YES -TBE ELISA and its cross reaction has been used in IFA positive samples, which is why we describe this method as follows.  After IFA method, all samples were verified by  ELISA and positive samples were confirmed for TBEV (Strain Hypr), WNV (WNV Strain Line 2), and USUV by a virus neutralization test (VNT) in micromodification, with vital staining (7 CV-1 cell suspension, monkey kidney cell line) used as a cell substrate for WNV and USUV, and suspension of PS cells (porcine kidney cell line) used as a cell substrate for TBEV, with a working dilution of 600,000 cells/ml for both cell lines. The result of VNT is a virus neutralization (VN) titer, which is the reciprocal of the highest sample dilution that is still capable of neutralizing the cytopathic effect at more than 50% (TCID50). Samples were marked as positive if the VN titer > 4

 Also why only one VNT sample has been proved positive is discussed in the line 191 and onwards

Results:

  • The table presented is not clear and contains errors. Because, by VNT the neutralizing immune response was considered specific if the VNT titer for any given virus was at least fourfold higher than titers obtained against the other viruses. And if the antibody titers for both viruses (WNV and USUV) were the same or differed only slightly (1 to 1,5 fold), it is not possible to discriminate between WNV or USUV titers and the result must be interpreted as not analyzable. Therefore e.g. for a pelican, a barn owl, a snowy owl, a wolf and a red fox the exact specific antibody reaction cannot be determined at the same time. Hence the presentation of the positive results must be revised. Or what are your comments when you have equal serological titers for WNV and USUV in same birds and mammals? What is with cross-reactions by the IFA and their results?

All IFA positive results have been confirmed by TBE EIA Elisa and following VNT specific tests for each of three possible pathogenes (TBE, WNV , USUTU ) as follows. All ELISA-positive samples were confirmed for TBEV (Strain Hypr), WNV (WNV Strain Line 2), and USUV by a virus neutralization test (VNT) in micromodification, with vital staining (7 CV-1 cell suspension, monkey kidney cell line) used as a cell substrate for WNV and USUV, and suspension of PS cells (porcine kidney cell line) used as a cell substrate for TBEV, with a working dilution of 600,000 cells/ml for both cell lines. The result of VNT is a virus neutralization (VN) titer, which is the reciprocal of the highest sample dilution that is still capable of neutralizing the cytopathic effect at more than 50% (TCID50). Samples were marked as positive if the VN titer > 4

Also why only one VNT sample has been proved positive is discussed in the line 191 and onwards

Discussion:

  • The part is quite well done and ties into many other results from zoo studies already.
  • The discussion should be adapted to the revised serological results and it would also be recommended to insert the zoo study from France (Constant et al. 2020)

The study from France (Constant et al. 2020) has been included in the discussion.

Finally, the results of the study should be conveyed in a more attractive style with informative tables and figures of the results for a better character of this longitudinal study of zoo animals over 17 years as an important factor that zoological collections can use for early detection of emerging diseases.

Combination of short table with the supplemental data seemed like a best fit for this manuscript in our opinion. I am not a fond of long tables which make data unclear. If we would like to present all animals species or animals divided according to the year od sampling, we think the tables would be too long and unclear.

Furthermore, after a major revision of all critical points and minding the fact that only a few methods were performed and the novelty of the results is limited, it should be rethought to publish this manuscript as a short communication form.

If that would be possible, we would like to keep the standard format for this manuscript.

Minor points: In the supplemental part each scientific name of the bird and mammal species should be written in italics.

Supplemental part has been corrected and all scientific names have been written in italics.

Reviewer 2 Report

In this manuscript by Kvapil et al., authors demonstrated an early detection method of circulation of West Nile Virus and Usutu virus in Slovenia.

The article is well written with detailed introduction and discussion section. Although the manuscript is one aimed study to serologically detect presence or absence of West Nile Virus and Usutu virus in zoo animals, in my opinion the idea is novel and has potential for great implication in early detection of these viruses circulation in a specific area.

I think this manuscript has high importance in surveillance studies and it is appropriate for publication in ‘Viruses’ in its’ present form.

Author Response

Cover letter

A Sentinel Serological Study in Selected Zoo Animals to Assess Early Detection of West Nile and Usutu Virus Circulation in Slovenia

Dear editor,

Thank You very much for Your review and positive outcome.

Pavel Kvapil

Reviewer 2

In this manuscript by Kvapil et al., authors demonstrated an early detection method of circulation of West Nile Virus and Usutu virus in Slovenia.

The article is well written with detailed introduction and discussion section. Although the manuscript is one aimed study to serologically detect presence or absence of West Nile Virus and Usutu virus in zoo animals, in my opinion the idea is novel and has potential for great implication in early detection of these viruses circulation in a specific area.

I think this manuscript has high importance in surveillance studies and it is appropriate for publication in ‘Viruses’ in its’ present form.

Reviewer 3 Report

To the authors. Please note I have attached the submitted pdf manuscript, with suggestions for many minor changes to be made (I used sticky notes).  Additional comments refer to the scientific content and detail (see below).

This manuscript describes the results of serological tests on sera derived from a variety of animal species (including birds) that have been maintained in healthy condition and in a protected environment, ie a zoo. The authors argue that the choice of zoological samples rather than from the wild, has the advantage of being able to monitor the immunological status of the animals over a period of several years.

The serological tests focused on the detection of antibodies against West Nile virus and/or Usutu virus using indirect immunofluorescence tests (primarily detecting IgG responses) and a virus neutralization test which was a modified version of the plaque reduction neutralisation test (PRNT) and, in my opinion, less sensitive and less precise than the plaque reduction neutralization test (PRNT).  Nevertheless, WNV and USUV still appear to be increasing their epidemiological and epizootic significance in Europe and are therefore of continuing interest to the virological community in Europe.

The manuscript is relatively short, moderately well written and most of the interpretation of the results is acceptable. However, I have taken the liberty of using “sticky notes” on the original pdf to draw attention to the numerous minor grammatical, typographical etc., points that require attention.  This corrected version of the manuscript is attached (for the authors to consider), together with the following comments.

First sentence of introduction. “Comment by this reviewer” - these viruses are already recognized zoonotic viruses.  Their potential is to be epizootic and even epidemic in Europe - therefore, you could consider re-writing the first sentence as follow;

“West Nile virus (WNV) and Usutu virus (USUV), in the family Flaviviridae, are emerging enzootic pathogens with epizootic and epidemic potential in Europe”.

Page 2 line 3.  The first cases of WNV in New York (in birds, horses and humans) were reported in August/Sept 1999 - not in 2002!  The authors appear to be mixing up the dates of the paper that is quoted (ie 2002) with the dates during which WNV was first identified as being present in birds, horses and humans in New York (in the Zoo! and elsewhere in the New York area).  The first paper referring to WNV in a Flamingo, held in the Zoo in New York was published in 1999 not 2002. (DOI:10.1126/science.286.5448.2333)

Page 1   On line 3 of the introduction, it correctly states that birds may act as a reservoir of the virus. Then it states “and in recent years, also birds can be infected”.  This is misleading. When birds are acting as a reservoir for the virus (either WNV or USUV) they must be infected, otherwise how do they carry the virus?  From the text it is not clear that when birds are the reservoir for the virus, they usually appear healthy but they are infected and therefore carrying the virus, perhaps even for long distances to new geographic regions during migration. On the other hand, birds that become infected may develop symptoms arising from the infection. As a general rule, these symptomatic birds have probably experienced the virus for the first time, for example, when healthy migratory birds introduce the virus to a region where the virus has not been circulating (or circulating at a low level) previously. Some of these latter birds may be quite young and still carrying maternal antibody/immunity which might protect them from the introduced virus. In other birds the maternal antibody may have declined or there may not have been any maternal antibody. These resident birds (for example in central or northern Europe) might be expected to be susceptible to infection by WNV and/or USUV.  Now consider this comment immediately below.

Page 1. On line four the sentence commences with the words “Incidental hosts…..” and it includes the words - “and in recent years also birds can be infected, resulting in febrile illness”. These sentences could be very confusing to readers unfamiliar with these viruses and their interactions with birds……  Also, birds cannot be reservoir hosts in one region of Europe and incidental hosts, in another region of Europe, since birds are the primary source of bloodmeals for the many ornithophilic Culex species mosquitoes in all regions!  I do understand what the authors are trying to say but it needs to be written more carefully to remove these apparent contradictions.

Page 2 – Penultimate line.   The authors state “sample dilution that is still capable of neutralizing the cytopathic effect in more than 50%....”  The words-  “in more than”  -give the impression that this is an arbitrary level of inhibition of cpe, above 50%, ie in some wells it could be 60% whereas in other wells it could be 90% or even 100%.  How is this variable level of inhibition estimated?  I note that a vital staining procedure is included in the VNT.  Does this mean that each well is monitored automatically, using an instrument that can detect differences in percentage of vital stain?  If it does, the appropriate details should be included in the Methods.  Alternatively, if the percentage of vital staining is estimated by eye, this should be admitted and the words “in more than” should be changed to “in at least 50% of each monolayer”.

Page 4 – in the key under Table 2 the authors refer to “Green shading”  - there was no green shading on the Table submitted for review.

Page 4 – In Table 1, the barn owl result shows as “1:128” by VNT for WNV but “1:129” for USUV this is obviously a typographical error and needs to be corrected. 

These positive VNT results (i.e. for the barn owl) are really difficult to understand, firstly because the corresponding IFA results were negative! This seems quite extraordinary. How is it possible to have strongly neutralizing sera (ie a functional test) that do not show up as positive in an indirect immunofluorescence test (ie, an antibody binding test)? It is clearly not due to the failure of the anti-species conjugate because serum, from the following year, is positive in the IFA test.  Were these IFA and VNT tests repeated and were they reproducible?  How many replicate wells were used to estimate the endpoint dilution in the VNT?  And were control negative viruses included in the analyses? Also, in the context of control negative viruses, TBEV is mentioned in the Materials and Methods but no experiments/results are presented for this virus. Was TBEV considered to be a likely negative control virus for all the tests or why was it included? Bearing in mind that in Europe, TBEV is predominantly a forest-associated tick-borne flavivirus and is not normally associated with Culex species associated mosquitoes what was the scientific rationale for including TBEV in the analysis of sera collected in the “protective environment” of a zoo?

Also, the authors should attempt to explain how it is possible for the VNT to produce quite a high titre result (for the barn owl serum) but the same serum was negative in the IFA test (also see my comments/suggestions later). 

Page 5  line 4 et seq - This section could be more informative. “Usutu virus was first discovered in Africa, in 1959 and subsequently for the first time on record, in southern Europe (Austria) in 2001, before gradually dispersing throughout many parts of mainland Europe. Moreover, WNV and USUV-specific neutralizing antibodies were detected in a wide range of healthy wild migratory and non-migratory birds, and sentinel chickens, in England as reported in 2003 and 2006 (references 22 and 23).  And an African lineage of Usutu virus has now been isolated from birds in the UK in 2020 (Ref PMID: 33063656). It is assumed that this USUV was introduced from Belgium or Holland where the same strain has been detected recently.

Page 5 – referring to the VNT titre of 1:128 for both viruses.   I understand the rationale behind the thinking, ie that both viruses may have induced high immunity levels.  However, this does not appear to be supported by the negative IFA results for the same sera and the relatively low level of immunity (by IFA) during a subsequent  (4 Dec 2017) analysis (as shown in Table 1).  Nevertheless, I have thought long and hard about this apparently anomalous result and a possible explanation could be that this owl had recently” been infected by WNV and/or USUV and had responded strongly in the VNT.  AND, if it was a recent infection, it would be a predominantly IgM response which would not have been readily detectable in the IFA test (which uses an IgG-specific conjugate) but could still have been detectable in the neutralization test!  I cannot think of any other reason for theis apparently anomalous result, unless the result is mistakenly presented for the 4th  December 2017 sample when it should have been presented for the 18th September 2018 sample?

In the Discussion, it may be worth considering why IFA  tests were positive but only one VNT test produced a positive result.  One possibility might be that since these animals have benefitted from being kept under the protective environment of the zoo, including the regular provision of food, protection from predators and climatically harsh conditions, etc., and, as they also have records of having been healthy for years, they clearly did not suffer severe infections due to WNV or USUV.  This might explain why low levels of virus-specific antibody could be detected in their sera (ie due to mild or asymptomatic infections) but only in one case was evidence of a potent neutralizing response (and therefore maybe a symptomatic infection occurred) but in the protective environment of the zoo the bird was sufficiently robust to recover from the infection).

Author Response

Cover letter

A Sentinel Serological Study in Selected Zoo Animals to Assess Early Detection of West Nile and Usutu Virus Circulation in Slovenia

Dear editor,

Thank You very much for Your thorough review and carefull corrections regarding grammar and spelling, I do appreciate Your time and patience.

Kind regards

Pavel Kvapil

Please find the rseponse bellow

Reviewer 3

To the authors. Please note I have attached the submitted pdf manuscript, with suggestions for many minor changes to be made (I used sticky notes).  Additional comments refer to the scientific content and detail (see below).

This manuscript describes the results of serological tests on sera derived from a variety of animal species (including birds) that have been maintained in healthy condition and in a protected environment, ie a zoo. The authors argue that the choice of zoological samples rather than from the wild, has the advantage of being able to monitor the immunological status of the animals over a period of several years.

The serological tests focused on the detection of antibodies against West Nile virus and/or Usutu virus using indirect immunofluorescence tests (primarily detecting IgG responses) and a virus neutralization test which was a modified version of the plaque reduction neutralisation test (PRNT) and, in my opinion, less sensitive and less precise than the plaque reduction neutralization test (PRNT).  Nevertheless, WNV and USUV still appear to be increasing their epidemiological and epizootic significance in Europe and are therefore of continuing interest to the virological community in Europe.

The manuscript is relatively short, moderately well written and most of the interpretation of the results is acceptable. However, I have taken the liberty of using “sticky notes” on the original pdf to draw attention to the numerous minor grammatical, typographical etc., points that require attention.  This corrected version of the manuscript is attached (for the authors to consider), together with the following comments.

First sentence of introduction. “Comment by this reviewer” - these viruses are already recognized zoonotic viruses.  Their potential is to be epizootic and even epidemic in Europe - therefore, you could consider re-writing the first sentence as follow;

Reviewer 3

  1. All sticky notes have been corrected., In the table all UV And eventually WN were corrected to proper abbreviations.

Line 166-167 year 2010 corrected to 2001, proper reference added

Line 170- reference added

  1. Content remarks.

Line-

Introduction-

First sentence of introduction. “Comment by this reviewer” - these viruses are already recognized zoonotic viruses.  Their potential is to be epizootic and even epidemic in Europe - therefore, you could consider re-writing the first sentence as follow;“West Nile virus (WNV) and Usutu virus (USUV), in the family Flaviviridae, are emerging enzootic pathogens with epizootic and epidemic potential in Europe”.

Has been corrected according to the suggestion.

Page 2 line 3.  The first cases of WNV in New York (in birds, horses and humans) were reported in August/Sept 1999 - not in 2002!  The authors appear to be mixing up the dates of the paper that is quoted (ie 2002) with the dates during which WNV was first identified as being present in birds, horses and humans in New York (in the Zoo! and elsewhere in the New York area).  The first paper referring to WNV in a Flamingo, held in the Zoo in New York was published in 1999 not 2002. (DOI:10.1126/science.286.5448.2333)

Line 53- date 2001 has been changed to 1999, I hope I understood Your remark correctly. Lanciotti referece from 1999 is number 7

Page 2 – Penultimate line.   The authors state “sample dilution that is still capable of neutralizing the cytopathic effect in more than 50%....”  The words- “in more than”  -give the impression that this is an arbitrary level of inhibition of cpe, above 50%, ie in some wells it could be 60% whereas in other wells it could be 90% or even 100%.  How is this variable level of inhibition estimated?  I note that a vital staining procedure is included in the VNT.  Does this mean that each well is monitored automatically, using an instrument that can detect differences in percentage of vital stain?  If it does, the appropriate details should be included in the Methods.  Alternatively, if the percentage of vital staining is estimated by eye, this should be admitted and the words “in more than” should be changed to “in at least 50% of each monolayer”.

Sentence “in at least 50% of each monolayer”. Has been changed accordingly

Page 4 – in the key under Table 2 the authors refer to “Green shading”  - there was no green shading on the Table submitted for review.

Green shading was erased due to the format of the journal and ther reference under the table has been erased

Page 4 – In Table 1, the barn owl result shows as “1:128” by VNT for WNV but “1:129” for USUV this is obviously a typographical error and needs to be corrected. 

Has been corrected

These positive VNT results (i.e. for the barn owl) are really difficult to understand, firstly because the corresponding IFA results were negative! This seems quite extraordinary. How is it possible to have strongly neutralizing sera (ie a functional test) that do not show up as positive in an indirect immunofluorescence test (ie, an antibody binding test)? It is clearly not due to the failure of the anti-species conjugate because serum, from the following year, is positive in the IFA test.  Were these IFA and VNT tests repeated and were they reproducible?  How many replicate wells were used to estimate the endpoint dilution in the VNT?  And were control negative viruses included in the analyses? Also, in the context of control negative viruses, TBEV is mentioned in the Materials and Methods but no experiments/results are presented for this virus. Was TBEV considered to be a likely negative control virus for all the tests or why was it included? Bearing in mind that in Europe, TBEV is predominantly a forest-associated tick-borne flavivirus and is not normally associated with Culex species associated mosquitoes what was the scientific rationale for including TBEV in the analysis of sera collected in the “protective environment” of a zoo?

This problem has occurred as the positive titres have been added to the wrong date during the formatting of the table.  The date has been corrected. Control in the other lab for the date December 2018 was ELISA TBEV which cross-react with USUV and WNV. Positive ELISA has been confirmed with VNT. To the same titres. This has been described in the original manuscript, but showed up a little confusing. During the editing, the date has been unfortunately misplaced. This corresponds with description in discussion, when correct time frame (autumn 2018 ) is mentioned.

Also, the authors should attempt to explain how it is possible for the VNT to produce quite a high titre result (for the barn owl serum) but the same serum was negative in the IFA test (also see my comments/suggestions later). 

Positive titres for VNT in Barn owl have been misplaced during the formatting of the table and date has been corrected.

Page 5  line 4 et seq - This section could be more informative. “Usutu virus was first discovered in Africa, in 1959 and subsequently for the first time on record, in southern Europe (Austria) in 2001, before gradually dispersing throughout many parts of mainland Europe. Moreover, WNV and USUV-specific neutralizing antibodies were detected in a wide range of healthy wild migratory and non-migratory birds, and sentinel chickens, in England as reported in 2003 and 2006 (references 22 and 23).  And an African lineage of Usutu virus has now been isolated from birds in the UK in 2020 (Ref PMID: 33063656). It is assumed that this USUV was introduced from Belgium or Holland where the same strain has been detected recently.

The section has been widened according to the suggestion.

Page 5 – referring to the VNT titre of 1:128 for both viruses.   I understand the rationale behind the thinking, ie that both viruses may have induced high immunity levels.  However, this does not appear to be supported by the negative IFA results for the same sera and the relatively low level of immunity (by IFA) during a subsequent  (4 Dec 2017) analysis (as shown in Table 1).  Nevertheless, I have thought long and hard about this apparently anomalous result and a possible explanation could be that this owl had “recently” been infected by WNV and/or USUV and had responded strongly in the VNT.  AND, if it was a recent infection, it would be a predominantly IgM response which would not have been readily detectable in the IFA test (which uses an IgG-specific conjugate) but could still have been detectable in the neutralization test!  I cannot think of any other reason for theis apparently anomalous result, unless the result is mistakenly presented for the 4th  December 2017 sample when it should have been presented for the 18th September 2018 sample?

The first option You have mentioned is an interesting possibility , but unfortunately , the second option has been proven right and the date was misdisplaced and corrected, section in discussion has been also widened according to the suggestion.

In the Discussion, it may be worth considering why IFA  tests were positive but only one VNT test produced a positive result.  One possibility might be that since these animals have benefitted from being kept under the protective environment of the zoo, including the regular provision of food, protection from predators and climatically harsh conditions, etc., and, as they also have records of having been healthy for years, they clearly did not suffer severe infections due to WNV or USUV.  This might explain why low levels of virus-specific antibody could be detected in their sera (ie due to mild or asymptomatic infections) but only in one case was evidence of a potent neutralizing response (and therefore maybe a symptomatic infection occurred) but in the protective environment of the zoo the bird was sufficiently robust to recover from the infection).

 This section has been widened according to the suggestion.

With kind regards

Pavel Kvapil

Round 2

Reviewer 1 Report

Dear Authors,

see my comments in the document in blue.

Author Response

Dear Reviewer, 

 Thank You for Your time and patience. Please find the response in the attachment as well as revised manuscript.

With kind regards

PK
